# Chitosan–Cu Catalyzed Novel Ferrocenated Spiropyrrolidines: Green Synthesis, Single Crystal X-ray Diffraction, Hirshfeld Surface and Antibacterial Studies

**DOI:** 10.3390/polym15020429

**Published:** 2023-01-13

**Authors:** Mohammad Asad, Muhammad Nadeem Arshad, Abdullah M. Asiri, Mohammed M. Rahman, Snigdha Kumaran, Mohammed Musthafa Thorakkattil Neerankuzhiyil

**Affiliations:** 1Chemistry Department, Faculty of Science, King Abdulaziz University, P.O. Box 80203, Jeddauh 21589, Saudi Arabia; 2Center of Excellence for Advanced Materials Research (CEAMR), King Abdulaziz University, P.O. Box 80203, Jeddah 21589, Saudi Arabia; 3Research & Postgraduate Department of Chemistry, MES Kalladi College, Mannarkkad 678583, India

**Keywords:** green synthesis, spiropyrrolidines, X-ray crystallography, Hirshfeld surface analysis, antibacterial activity

## Abstract

Chitosan-bounded copper (chitosan–Cu) was introduced for green synthesis of novel ferrocenated spiropyrrolidine hybrids, namely 3′-(4-.bromobenzoyl)-5′-(4-hydroxybenzyl)-4′-ferrocenylspiro[indoline-3,2′-pyrrolidin]-2-one and 3′-(4-bromobenzoyl)-4′-ferrocenylspiro[indoline-3,2′-pyrrolidin]-2-one, in good yield. A one-pot three-component 1,3-dipolar cycloaddition reaction was employed for the formation of spiropyrrolidines from 1-(4-bromophenyl)-ferrocene-prop-2-en-1-one and azomethine ylides, which were developed in situ from tyrosine, glycine, and isatin, respectively. Various spectroscopic methods were used to establish the structures of spiropyrrolidines, and a single crystal X-ray diffraction study of a spiropyrrolidine provided additional confirmation. The crystallographic study revealed that compound **3a** has one independent molecule in its unit cell, which is correlated with Hirshfeld surface analysis, and describes intramolecular contacts adversely. The highly yielded products in green conditions were determined for their antibacterial significance and were found to have good activity against Gram-positive and Gram-negative bacterial strains.

## 1. Introduction

Nitrogen-containing heterocyclic scaffolds are a significant class of compounds [1,2]. Many of heterocyclic scaffolds have been synthesized by intermolecular 1,3-dipolar cycloaddition reaction [3]. The simplest and most practical way to make five-membered spiroheterocycles is through this reaction. Pyrrolidines are an important class of *N*-containing heterocycles [4]. These five-membered spiroheterocycles have exhibited potential biological activity, including glucosidase inhibition, anticancer, antidiabetic, antiviral, antibacterial, and antimycobacterial activities [5,6,7,8,9,10]. Therefore, pyrrolidine and other nitrogen-containing moieties, such as pyrazoline, are interesting and many researchers have designed and targeted the synthesis of these heterocycles as potential analogues [11,12,13]. 

In a continuation of our work, we are also focusing on preparing organometallic-grafted spiroheterocycles. Organometallic compounds usually hold significant efficacy due to their functioning in various commercial areas [14]. The synthesis of ferrocene-based heterocycles has attracted new attention because of their potential uses. Hence, as a simple and less costly organometallic material, ferrocene has received special attention in recent ongoing research for biologically active organic molecule synthesis.

One of the main objectives of “Green Chemistry” is the development of energy-efficient protocols. As such, catalysis receives much more attention in green chemistry research. The sustainability of a transition metal-catalyzed process depends on the catalyst’s recovery and recycling, because metal catalysts are typically expensive. Recovery and recycling of the catalyst is crucial for the sustainability of a transition metal-catalyzed process, since metal catalysts are usually costly and harmful from both an economic and environmental standpoint [15]. Therefore, it has become a trend among researchers that catalytically active component’s immobilization on polymeric materials as plausible supports because they increase the catalyst’s selectivity, stability, and reusability. Recently, there has been a lot of interest in metal catalysts anchored on a heterogeneous support [16]. The benefits of heterogeneous processes include easier product separation, reusable catalyst, and improved reaction steric control intermediates. These advantages have prompted researchers to immobilize a homogeneous catalytic site on a range of supports, including silicates, zeolites, magnetic materials, and polymers [17,18,19,20]. Due to its environment-friendly properties and crucial functions in transition metal-catalyzed processes, chitosan (CS) has recently attracted a lot of interest [21]. Chitosan is a reflective biopolymer due to its non-toxic, biodegradable, biocompatible, chemo-physical, hydrophilic positive charged, and biological characteristics. It is a suitable heterogeneous catalyst and catalytic supporting medium due to its free –NH_2_ groups and hydrogen bonding.

Chitosan–Cu catalyst was obtained by the immobilization of Cu on chitosan [22]. The presence of free amino groups makes chitosan a suitable solid support [22]. The polymer-immobilized Cu complexes were reported to be more stable for a wide range of pH values [23] and were used as catalyst for the C−N bond formation reaction [24], C−S coupling reaction [22], cyclopropanation reactions [25], etc. The reason may be that the linkage to a polymeric material, such as chitosan, provides a more constraining environment for metal chelation than that of polymer-free ligands [25]. Moreover, compared to transition metal catalysts of Cu, the chitosan-supported catalysts are insoluble in most solvents, and the catalysts can be easily recovered after the reaction by simple filtration from reaction mixture, meaning that they are reusable for next several cycles; they are also are good candidates as biocompatible catalysts [25].

For advancement, 1,3-dipolar cycloaddition processes have been performed utilizing microwave irradiation [26], ultrasonic irradiation [27], and ionic liquid [28] employing azomethine ylide. Herein we report the ferrocene-grafted spiropyrrolidine hybrids, which were synthesized by 1-(4-bromophenyl)-ferrocene-prop-2-en-1-one using a 1,3-dipolar cycloaddition reaction as a dipolarophile with different azomethine ylides via solvent-free reaction using chitosan–Cu as a green catalyst [22]. The spectroscopic techniques were used for establishing the structures of synthesized spiropyrrolidines, and further validated by single crystal X-ray diffraction study, which was additionally combined with Hirshfeld surface analysis. The new synthesized spiropyrrolidine was determined for antibacterial significance, and its efficacy was compared with a standard drug.

## 2. Materials and Methods 

### 2.1. General

All the chemicals were used as acquired. All of the chemicals consumed were purchased from an international chemical supplier. The melting point was determined using a Stuart Scientific SMP3, version 5.0, melting point equipment (Bibby Scientific Limited, Staffordshire, UK), which was reported as uncorrected, on a clean Thermo Scientific NICOLE iS50 FT-IR spectrometer (Madison, WI, USA). Using TMS as an internal standard, the ^1^H, and ^13^C spectra were captured using Bruker 850 MHz equipment in a CDCl_3_. Chemical shifts are displayed in ppm, while coupling constants are given in hertz. Aluminum sheets covered with silica gel (Type 60 GF254, Merck, Darmstadt, Germany) were used for TLC, and the spots were found by exposing the sheets to UV light at either 254 or 360 nanometers. A PerkinElmer 2400 Series II Elemental CHN analyzer was used to perform elemental analyses. MoKa (k = 0.71073A) radiation was used to collect the single crystal X-ray data set for 2a on the Bruker Kappa X8 APPEX II diffractometer. The scan area was 2.22° to 33.00°.

### 2.2. Synthesis of Catalyst, Chitosan–Cu 

Chitosan–Cu catalyst has been synthesized according to the reported procedure with minor modifications [22]. Here, Cu(CH_3_CO_2_)_2_ (200 mg) was dissolved in water and added to a suspension of chitosan (1 g) in 20 mL of water, which was then agitated for 3 h. After the copper had been adsorbed, the solid was filtered and extensively cleaned with water to remove lingering Cu compounds. It was then dried under vacuum for a whole night at 50 °C to produce the chitosan–Cu catalyst.

### 2.3. Synthesis of Spiropyyrolidines *(**3a**–**b**)*

#### 2.3.1. General Procedure for Conventional Synthesis of Spiropyyrolidines (**3a**–**b**)

Chalcone (1 mmol), isatin (0.147 g, 1 mmol), and amino acids (1 mmol) were mixed and dissolved in 10 mL of methanol. The mixture was then refluxed for 5 h. The reaction mixture was kept at room temperature for 30 min and poured in to 50 mL ice cold water. We filtered the precipitate, washed with water, and let it dry and recrystallize in chloroform–methanol to result in the pure **3a**–**b** samples.

#### 2.3.2. Green Method for Synthesis of Spiropyyrolidines (**3a**–**b**)

Isatin (0.147 g, 1 mmol), chalcone (1 mmol), and amino acids (1 mmol) were mixed well with the catalyst chitosan–Cu (50 mg) in a mortar. We transferred the reaction mixture into a 100 mL beaker, and it was heated at 60 °C. The reaction progress was checked by TLC. After the reaction completion, hot ethyl acetate was poured to recover the insoluble catalyst, which was then recovered by filtration. The filtrate was evaporated, and crude products **3a** and **3b** were recrystallized from the analytical grade chloroform–methanol mixture. 

#### 2.3.3. Synthesis of Spiropyyrolidines under Different Solvent Assisted Condition Using Chitosan–Cu Catalyst

Chalcone (1 mmol), isatin (0.147 g, 1 mmol), and tyrosine (1 mmol) mixture were dissolved in diverse solvents (chloroform, PEG-400, methanol, and H_2_O) (10 mL) and catalyst (50 mg) was added to it. The reaction mixture was refluxed for a particular time duration. The reaction progress was checked by TLC. After completion of the reaction the synthesized product was isolated and purified from the reaction medium by adding hot ethyl acetate. The catalyst was collected using filtration, and the remaining solvent was evaporated. A chloroform–methanol mixture was used to recrystallize the mixture to produce compound **3a**.

#### 2.3.4. Synthesis of Spiropyyrolidines under Solvent-Free Condition Using Different Catalytic System

A mixture of chalcone (1 mmol), isatin (0.147 g, 1 mmol), and tyrosine (1 mmol) were mixed well with different catalytic systems, namely sodium acetate, copper acetate, CuSO_4_, and chitosan, respectively, in a mortar, separately. We transferred the reaction mixture into a 100 mL beaker, and it was heated at 60 °C. Then, TLC was used to check the reaction progress After the reaction completion, ethyl acetate was added for the recovery of the insoluble catalyst. The filtrate was evaporated, and the chloroform–methanol was used to recrystallize the pure compound **3a**.

#### 2.3.5. 3′-(4-Bromobenzoyl)-5′-(4-hydroxybenzyl)-4′-ferrocenylspiro[indoline-3,2′-pyrrolidin]-2 One (**3a**)

Obtained as light yellow crystals; yield 72%; m.p. 148–150 °C; IR (KBr) values are as follows: ν_max_—3415, 3308, 2944, 1710, and 1668 cm^−1^; ^1^H NMR(CDCl_3_) δ_H_ 1.27 (s, 1H), 1.61 (s, 1H), 2.30 (s, 1H), 4.21 (t, *J* = 7.6, 1H), 4.53 (t, *J* = 1.7 Hz, 1H), 4.62 (t, *J* = 1.7 Hz, 1H), 4.68 (s, 1H), 4.70 (s, 1H), 4.72–4.76 (m, 6H), 4.80 (d, *J* = 7.6 Hz, 1H), 6.81 (d, *J* = 8.5, 1H), 6.84–6.95 (m, 2H), 7.10 (t, *J* = 7.6, 1H), 7.44 (d, *J* = 8.6, 1H), 7.55 (d, 9.3, 1H), 7.65–7.66 (m, 4H), 7.78–7.80 (m, 2H), 7.96 (s, 1H), and 8.10 (s, 1H) ppm. ^13^C NMR (CDCl_3_) values are as follows: δ_C_, 29.7, 34.8, 40.2, 58.3, 61.6, 67.0, 67.3, 68.0, 69.1, 69.8, 71.6, 118.4, 122.3, 123.1, 123.6, 125.2, 125.6, 127.4, 127.9, 128.3, 129.1, 129.9, 130.2, 131.8, 133.8, 140.0, 137.3, 147.5, 188.5, and 194.2 ppm; analysis calculated for C_35_H_29_BrFeN_2_O_3_ is as follows: C, 63.56; H, 4.42; N, 4.24; found—C, 63.37; H, 4.50; N, 4.33. 

#### 2.3.6. 3′-(4-Bromobenzoyl)-4′-ferrocenylspiro[indoline-3,2′-pyrrolidin]-2-one (**3b**)

Obtained as brown crystals; yield 78%; mp. 168–169 °C); IR (KBr) is as follows: ν_max_—3042, 1710, 1672, and 645 cm^−1^; ^1^H NMR (CDCl_3_, 850 MHz) are as follows: δ_H_ 2.80 (t, *J* = 8.5 Hz, 1H), 3.12 (t, *J* = 7.5 Hz, 1H), 3.94 (d, *J* = 8.50, 2H), 4.32 (s, 1H), 4.41 (s, 1H), 4.46 (s, 1H), 4.50–4.55 (m, 6H), 4.63 (d, *J* = 10.5 Hz, 1H), 6.35 (d, *J* = 6.5 Hz, 1H ), 6.85 (t, *J* = 8.6 Hz, 1H), 7.22 (t, *J* = 6.5 Hz, 1H), 7.4 (d, *J* = 7.6 Hz, 1H), 7.45-7.62 (m, 4H), and 8.10 (s, 1H) ppm; ^13^C NMR (CDCl_3_, 213.7 MHz) are as follows: δ_C_, 29.5, 36.2, 38.0, 58.4, 63.1, 65.8, 67.3, 67.8, 68.0, 69.2, 73.4, 114.3, 121.8, 123.6, 124.7, 125.5, 126.8, 127.4, 132.1, 132.9, 137.5, 181.2, and 196.3 ppm; mass is as follows: m/z = 554.03 (M+); analysis calculated for C_28_H_23_BrFeN_2_O_2_ is as follows: C, 60.57; H, 4.18; N, 5.05; found—C, 61.32; H, 4.46; N, 4.83.

### 2.4. Recycling and Reusage of Chitosan–Cu Catalyst

A mixture of chalcone (1 mmol), isatin (0.147 g, 1 mmol), and tyrosine (1 mmol) were mixed well with the catalyst chitosan–Cu (50 mg) in a mortar. We transferred the reaction mixture into a 100 mL beaker, and it was heated at 60 °C. The reaction progress was checked by TLC. Hot ethyl acetate added to recover the insoluble catalyst, and collected by filtration after the completion of reaction. We evaporated the filtrate, and the crude product **3a** was recrystallized from chloroform–methanol mixture. The residual catalyst was washed with an isopropyl alcohol–ethyl acetate (1:9) solution, dried, and then used for the next cycles.

### 2.5. Single Crystal X-ray Crystallography

The prepared material was added to the suitable solvent until the preparation of the concentrated solution. The slow evaporation methodology was used to obtain the suitable single crystal. The selected fine and well-shaped single crystal was fixed onto the assembly and position-fitted with microfocus Cu/Mo Kα radiation for data collection *via* the Agilent SuperNova Dual Source Diffractometer (Agilent Technologies). The CrysAisPro software [29] was used for data collection, the system temperature was 296 K, and Mo Kα radiation was the source of radiation. The structure was solved and further refined with SHELXS-97 following the direct approach method [30], and improved by utilizing full-matrix least-squares methods on *F*^2^, using SHELXL-97 [30]. The WinGX software [31] was used as a parent software to deal with structure solution and refinement. All non-hydrogen atoms were refined anisotropically using the full-matrix least squares techniques [27]. The required figures for understanding were generated with the use of well-acknowledged tools, such as *PLATON* [32] and *ORTEP* [33]. 

The *U*_iso_ was fixed at 1.2 times for the C(H) groups (aromatic, cp-ferrocene, and methine), C (H, H) groups (methylene), and for N(H) groups (amide and secondary amine). The *U*_iso_ was fixed at 1.5 times for the O(H) group (hydroxyl). All the available hydrogen atoms (C−H, N−H, O−H) were refined with riding coordinates. Crystal data were deposited at the Cambridge Crystallographic Data Centre; the following deposition number 2215469 was assigned to the compound’s title CCDC number. Crystal data is available at no cost upon request to the CCDC, 12 Union Road, Cambridge, CB21 EZ, United Kingdom; (Fax—(+44) 1223 336-033; email—data_request@ccdc.cam.ac.uk).

### 2.6. Antibacterial Study

The synthesized compounds (spiropyrolidines) were tested for their capacity to avert the projected growth of several bacterial strains, including *Pseudomonas aeruginosa* (*P. aeruginosa*), *Streptococcus pyogenes* (*S. pyogenes*), *Escherichia coli* (*E. coli*), *Klebsiella pneumoniae* (*K. pneumoniae*), and *Streptococcus aureus* (*S. aureus*) using the disk diffusion method [34,35,36]. The standard inoculums (1 − 2 × 107 cfu/mL 0.5 McFarland standards) were obtained and dispersed on the sterile agar plate surface. After being dry heat sterilized for 1 h at 140 °C, the 6 mm-diameter microbial discs were prepared for soaked and put onto an inoculation plate. The previously soaked discs in DMSO and ciprofloxacin (30 g) were used as positive and negative controls, respectively. By flipping the plate and incubating it at 37 °C for 24 h, the mixture’s impurities were removed. The antibacterial efficacy against several bacterial species was determined by the size of the inhibition zone using a broth microdilution technique. The tested compounds and standard drug were logarithmically and sequentially diluted two-fold in nutrient broth, and then growing bacterial cells at a concentration of about 5 × 10^5^ cfu/mL were injected into the bacterial culture. After 24 h of incubation at 37 °C, bacterial growth was measured visually and spectrophotometrically. The MIC refers to the minimal concentration of a drug necessary to inhibit bacterial growth. The 100 L bacterial culture-inoculated sterile agar plates were used to calculate the minimal bactericidal concentration (MBC). The CFU counts were calculated after an incubation period of 18 to 24 h at 35 °C. The minimum concentration of a compound required to kill 99.9% of inoculum is the MBC value.

## 3. Results and Discussion

Spiroheterocyclic scaffolds are generally synthesized by the dipolarophiles reacting with azomethine ylides. Moreover, chalcones instead of simple dipolarophiles are used for the spiropyrrolidines formation mentioned by few authors [37,38,39]. Previously, we have reported a ferrocenated chalcone as 1-(4-bromophenyl)-ferrocene-prop-2-en-1-one **1**, which was synthesized using the condensation of 4-bromoacetophenone and ferrocenecarboxaldehyde [40]. Herein, we report the novel spiropyrrolidine derivatives **3a**–**b** synthesis using a 1,3-dipolar cycloaddition reaction of chalcone **1** with azomethine ylides formed in situ from tyrosine, glycine, and isatin, respectively. The synthetic strategy of **3a**–**b** is depicted in Figure 1. 

Initially the reaction was carried out under the usual solvent-assisted refluxing conditions, with methanol serving as the solvent. The reaction was found to be finished in 5 h; however, there was no appreciable product yield at any parameter, and, moreover, the method was not environmentally friendly. This inspired us to use solvent-free heating method for the same reaction, and the reaction was performed by heating the starting ingredients in the presence of a heterogeneous catalyst, chitosan–Cu, at 60 °C. The catalyst was synthesized according to the reported literature with minor modifications [25]. Chitosan, a bio-based polymer, is highly stable in acidic as well as basic medium and is, therefore, able to form a good heterogeneous support. The immobilization of Cu on chitosan helps to increase the surface area, stability, increased number of active sites, etc. Thus, the combined effect of both the support and acidic catalytic system afforded **3a**–**b** in good yields (88–91%) in a shorter time span (Table 1). The obtained products were purified using the appropriate solvents, and the IR and NMR spectra were used to determine the structures of compounds. The additional structural confirmation was followed by a single crystal X-ray diffraction study. 

The reaction condition was optimized with further studies using a model reaction. The model reaction was selected from chalcone, isatin, and tyrosine, which yields **3a**. Initially, different kinds of solvents were selected for the examination of the solvent effect, namely methanol, PEG-400, chloroform, and water, along with the chitosan–Cu catalyst. A solvent-free reaction was also compared along with it. The comparative study shows that a yield of 75% was produced after 50 min when using methanol as a solvent. Similarly, PEG-400 took 1 h for completion of its reaction, with a 65% of yield. The nonpolar solvent chloroform resulted in a 64% yield after 1.5 h. A much better result in terms of yield (78%) was obtained in water compared to other solvents. However, solvent-free heating condition afforded **3a** a good yield (91%) in a shorter time span (15 min). Thus, the solvent-free condition became the efficient method in terms of relevant yield and time for this synthesis, as can be seen in Table 2.

To determine the superiority of chitosan–Cu as a catalyst, a model reaction was studied with different catalysts in solvent-free condition. For this purpose, sodium acetate, copper acetate, CuSO_4_, and chitosan were selected. The result obtained with Na(CH_3_CO_2_) was not satisfactory for the concerned reaction. Then, Cu(CH_3_CO_2_)_2_ and chitosan were evaluated separately. Cu(CH_3_CO_2_)_2_ resulted a 56% yield after 40 min. Similarly, the chitosan also resulted in same product in 1.5 h with a poor yield of 45%. Therefore, we expected that the combination of these two can produce much improved result in this reaction, and this become true, as shown in Entry 4 of Table 3. Chitosan–Cu can catalyze the said reaction much faster with an excellent yield of products. Immobilization of copper on chitosan increases the surface area of the catalyst and causes more active sites to become available for the interaction of reactants, enhancing the rate of reaction. 

Thus, the use of the chitosan–Cu catalyst in solvent-free condition was found to be the best condition for the synthesis of novel spiropyrrolidine derivatives **3a**–**b** using a 1,3-dipolar cycloaddition reaction of chalcone **1** with azomethine ylides formed in situ from tyrosine, glycine, and isatin, respectively. Moreover, this combination has the advantage of the insolubility of the catalyst, which facilitates easy separation and simplifies the procedure.

The reusability is one of the green synthesis criteria to be proven by a heterogeneous catalyst. The investigation into catalyst recycling was carried out; ethyl acetate was added to the reaction mixture at the completion of the reaction (confirmed by TLC). It only took a simple filtration to separate the catalyst solution from the products due to the high solubility of the product and the insolubility of the catalyst in ethyl acetate. The catalyst, which was left over as residue, was washed with an isopropyl alcohol–ethyl acetate (1:9) solution, dried, and utilized for the subsequent cycles. The recorded catalyst was effective for roughly five consecutive runs without seeing a material drop in catalytic activity (Table 4). 

The plausible mechanism for spiropyrrolidine **3a** formation has been given in Figure 2. The reaction starts with the activation of the carbonyl oxygen of the isatin by coordination with the chitosan–Cu catalyst and further nucleophilic attack of tyrosine. The intermediate formed undergoes dehydration and decarboxylation to produce another nucleophile which attacks the β-carbon of chalcone, followed by cyclization to result in spiropyrrolidine **3a** as the product. 

In the **3a–b** IR spectrum**,** the common absorption bands assigned at around 3310, 2923, 1718, and 1682 cm^−1^ correspond to NH, CH, and two C=O groups, respectively. The ^1^H NMR spectrum of **3a** exhibited two singlet peaks at 1.27 and 1.61 ppm assigned to tyrosine CH_2_ and OH protons, respectively. The triplet signals at 4.21, 4.53, and 4.62 ppm were assigned to three CH protons. Due to the ferrocene moiety, the two protons appeared as singlets at 4.68 and 4.70 ppm, the six protons appeared as a multiplet at 4.72–4.76 ppm, and one proton appeared as a doublet at 4.80 ppm. The 12 aromatic protons were assigned as doublets, triplets, and a multiplet in the region 6.80−7.81 ppm. The rest of the oxindole NH and tyrosine OH protons appeared as singlet 7.96 and 8.10 ppm, respectively. In the ^13^C spectrum, the two broad peaks at 188.5 and 194.2 ppm were assigned to two C=O carbons. The two CH and one CH_2_ carbons appeared at 34.8, 40.2 and 29.7 ppm, respectively.

### 3.1. Crystal Structure Description

Materials synthesis and their characterization is important in applied sciences for their applications in various aspects of life [41,42,43,44]. Therefore, this importance encouraged us to synthesize the heterocyclic materials and to perform their characterization by using well-equipped techniques, such as nuclear magnetic resonance (NMR) imaging and single crystal X-ray diffraction (SC-XRD). Compound **3a**, shown in Figure 1, is an organic substance with heterocyclic moieties in it. The central heterocyclic pyrrolidine ring (C1/C2/C3/C4/N1) is substituted by four different groups at each carbon atom. The 4-hydroxy benzyl ring is attached at carbon C3, while ferrocene (Cp-ring) is attached to the pyrrolidine ring at carbon C2. Carbon atom C1 of pyrrolidine is attached by the 4-bromobenzoyl group, while the carbon atom C4 is involved in spiro ring formation with an indolone system. The bond lengths for the bonds in the pyrrolidine ring are C1−C2 = 1.530(4) Å, C2−C3 = 1.540(4) Å, C3−N1 = 1.469(3) Å, N1−C4 = 1.463(4) Å, and C4−C1 = 1.567(4) Å. The N−H sites of the pyrrolidine and indolone systems are available for any substitution. The pyrrolidine ring is not fully planar with the root mean square (r.m.s) deviation of 0.1596, where maximum deviations were observed from the C1 = 0.2043 Å and C2 = −0.2229 Å. The fitted atoms of indolone rings (C4−C11/N2), however, are planar with the root mean square (r.m.s) deviations of 0.0323 Å. The dihedral angle between the planes of the indolone system and pyrrolidine ring is 85.68(10)° which indicates that both the planes are perpendicular to one other. The orientation angle between the plane, produced from its fitted atoms, of the 4-hydroxy benzyl ring (C30−C35) and pyrrolidine heterocycle system (C4−C11/N2), is 53.75(9)°. On the other side, the orientation of the benzoyl aromatic ring (C13−C18) and pyrrolidine ring (C4−C11/N2) is 78.65(2)°. This shows how the different planes of the same molecules are oriented with each other to stabilize the structure of the compound. In addition to this, the puckering parameters for the ring (C1/C2/C3/C4/N1) have been observed as *Q* and *φ*, with values of 0.357(3) Å and 82.8(5)°, respectively. The observed bond lengths (Å), bond angles (°), and torsion angles (°) are provided in the Appendix A, while the unique crystallographic parameters were stated in Table 5. The molecules undergo different C−H…O, N−H…O and O−H…O-type hydrogen bonding interactions to connect each other. The C15−H15…O1 interaction links geometrically to the molecules beside (010) to produce infinite extended chains. The N2−H2N…O3 interaction connects the molecules along (001) and generates infinite long chains. The strong O3−H1O…O2 hydrogen bonding interaction unites the molecules along (010) in a zig-zag fashion. The interactions N2−H2N…O3 and O3−H1O…O2 form a 12-membered ring motif which can be represented as $R_{4}^{4}\left(12\right)$ in Table 6, and Figure 2 and Appendix A [45]. Symmetry operations for the interactions N2−H2N…O3; C15-H15…O1 and O3−H1O…O2 are x, −y + 1/2, z−1/2; x, y + 1, z and −x + 2, y−1/2, and −z + 3/2, respectively, as in Table 6. All these interactions cause a two-dimensional (2D) network along the base vectors (0 1 0) and (0 0 1); parallel to the plane (100), as seen in Appendix A.

### 3.2. Hirshfeld Surface Analysis

The crystallographic file of the compound was studied more on Crystal Explorer [46] as a theoretical investigation into some useful interconnections. This is an approach in which a particular study endorsed the interactions which were observed in experimental crystallographic analysis. Figure 3 shows the different maps indicating the nature of contacts (a–f). The variation in colors, such as red, blue, and white, indicates the intensity or type of contacts from strong to weak. Appendix A, added to the Appendix A part, shows the hydrogen bonding interactions. The fingerprint plots of the compounds were generated and shown in Figure 4, indicating the specific amount of interaction in percentage terms from the certain sets of atoms. The H−H sets of atoms are contributing a major proportion of interactions (48.3%). For compound **3a**, the contribution of different bonds is as follows: H−C/C−H = 24.3%, O−H/H−O = 12.6%, H−Br/Br−H = 10.4%, C−C/C−C = 1.8%, C−Br/Br−C = 1.3%, N−C/C−N = 0.8%, and C−O/O−C = 0.2%. 

### 3.3. Antibacterial Activity

The antibacterial efficacy of the spiropyrrolidines **3a**–**b** against various Gram-positive and Gram-negative antibacterial pathogens, such as *Streptococcus pyogenes, Staphylococcus aureus, Klebsiella pneumoniae, Pseudomonas aeruginosa*, and *E. coli* was evaluated. The antibacterial activity findings of **3a**–**b** demonstrated moderate to good activity (Table 7). Compared to the effectiveness of the standard antibiotic ciprofloxacin, **3a** and **3b** were found to be more active against *S. pyogenes* and methicillin-resistant Staphlococcus aureus (MRSA+Ve) (MIC−25 g/mL) (Table 8). Additionally, studied have indicated that these compounds are efficient against *S. aureus*, *E. coli*, and *S. pyogenes*. In actuality, Gram-positive bacterial strains were more effectively combated than Gram-negative bacterial strains by both compounds. In comparison to the equivalent MIC results of **3a** and **3b** revealed by the disk diffusion method, the MBC of compounds was found to be two, three, or four times greater. The results of the antibacterial study revealed that compounds have moderate to good antibacterial activity. The MBC values for both compounds were discovered to be two, three or four times greater than the corresponding MIC values (Table 8). 

## 4. Conclusions

In conclusion, the present investigation results showed that chitosan–Cu catalyzed the green synthesis of spiropyrrolidine derivatives which have advanced superiority in terms of improved product and short reaction times over spiropyrrolidines which were conventionally synthesized. The FTIR and NMR spectral analyses has been established to investigate the newly synthesized spiropyrrolidine structures, which were moreover corroborated by the diffraction study (single crystal X-ray). The crystallographic study confirmed that compound **3a** has one independent molecule in its unit cell, which is correlated with Hirshfeld surface analysis, and described intramolecular contacts adversely. The good availability and dispersal of active chemical sites, which promote stronger interaction between substrate molecules and the catalyst, may be the cause of the key and maximum catalytic activity under solvent-free conditions. Additionally, because there is no solvent as a medium, there is no dilution effect, and the heat required for activation energy is directly available to the reactant molecules, which also help to improve yield as well as to reduce reaction time. Therefore, this methodology becomes greener, with attractive features, such as being eco-friendly, having a mild reaction condition, reduced period of reaction time, excellence in yields, recyclability of catalyst, etc. The target compound was crystalized in monoclinic crystal system with the space-group *P*2_1_/c. The unit cell dimensions are *a* = 16.7910(13) Å, *b* = 7.8298(6) Å, and *c* =23.1312(16) Å, while the beta angle is *β* = 106.316(8)°. The central pyrrolidine ring is substituted by various groups, and different functional groups are involved in hydrogen bonding interactions to provide stability to the structure. The spiropyrrolidines **3a–b** were afforded in high yields in green conditions, and were determined for their antibacterial significance and found to show good activity against Gram-positive (G^+^) and Gram-negative (G^−^) bacterial strains.

## Figures and Tables

**Figure 1 polymers-15-00429-f001:**
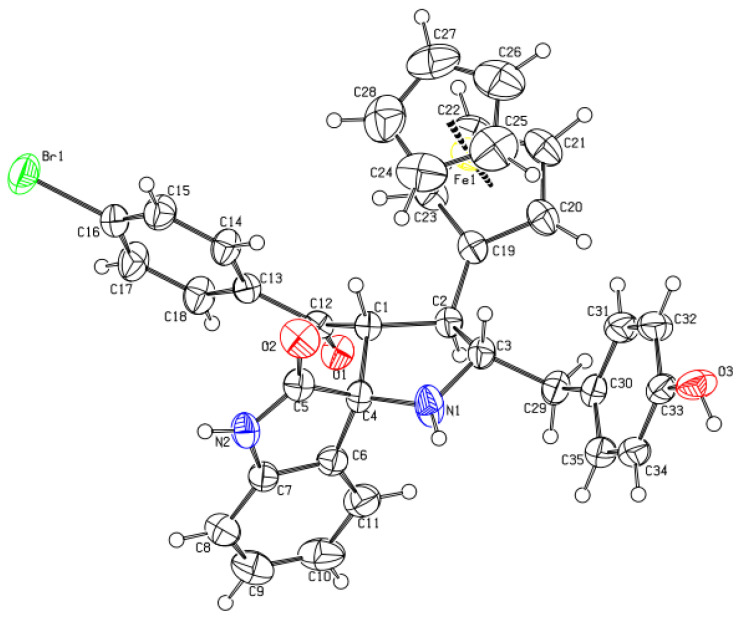
*ORTERP* diagram of **3a**, where thermal ellipsoids were drawn at 40%.

**Figure 2 polymers-15-00429-f002:**
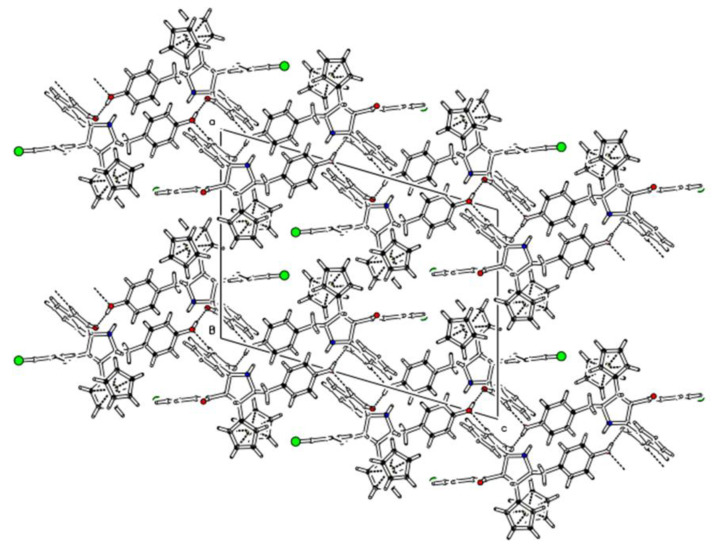
A view along the *ac*-plane showing different hydrogen bonding interactions.

**Figure 3 polymers-15-00429-f003:**
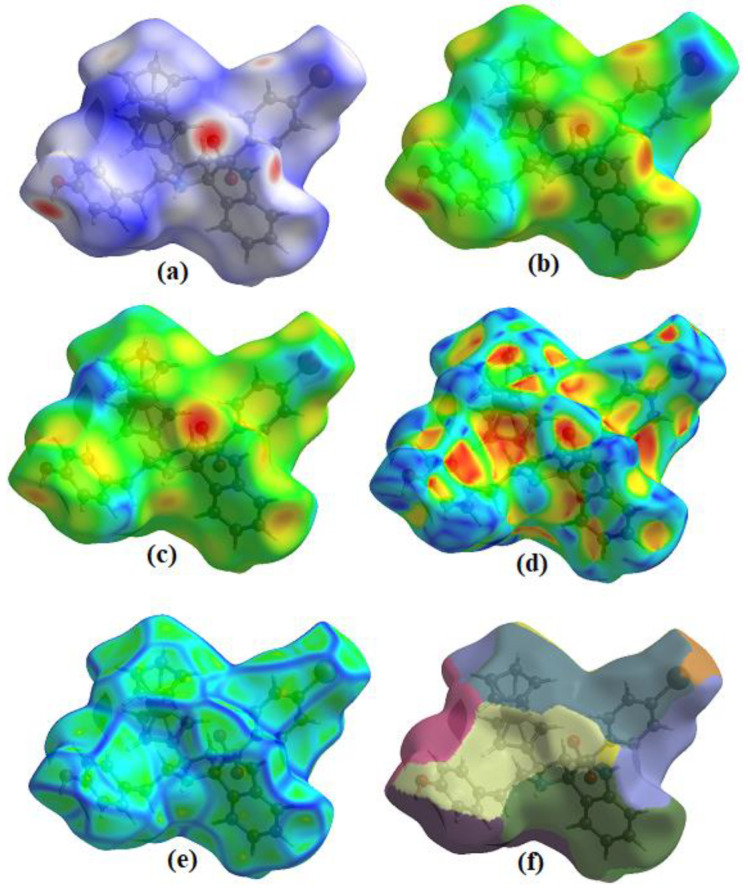
Hirshfeld maps for **3a** mapped on (**a**) *d*_norm_, (**b**) *d*_e_, (**c**) *d*_i_, (**d**) shape index, (**e**) curvedness, and (**f**) fragment patch.

**Figure 4 polymers-15-00429-f004:**
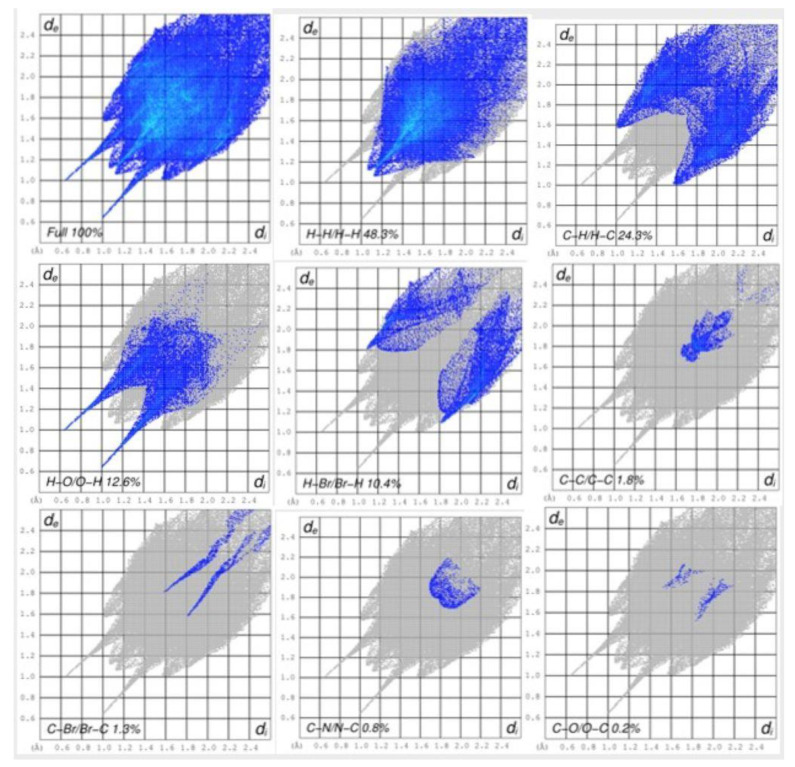
Full and partial 2D plots (fingerprint) for **3a**, showing the percentages of interatomic contacts.

**Scheme 1 polymers-15-00429-sch001:**
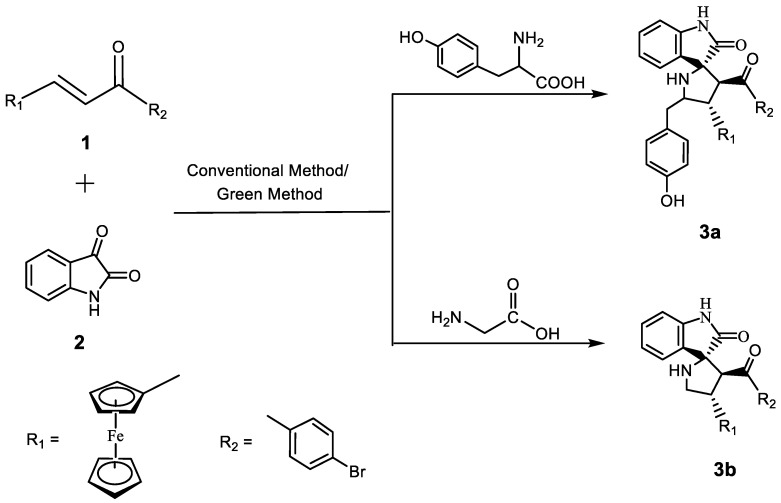
Synthesis of spiropyrrolidines **3a**–**b**.

**Scheme 2 polymers-15-00429-sch002:**
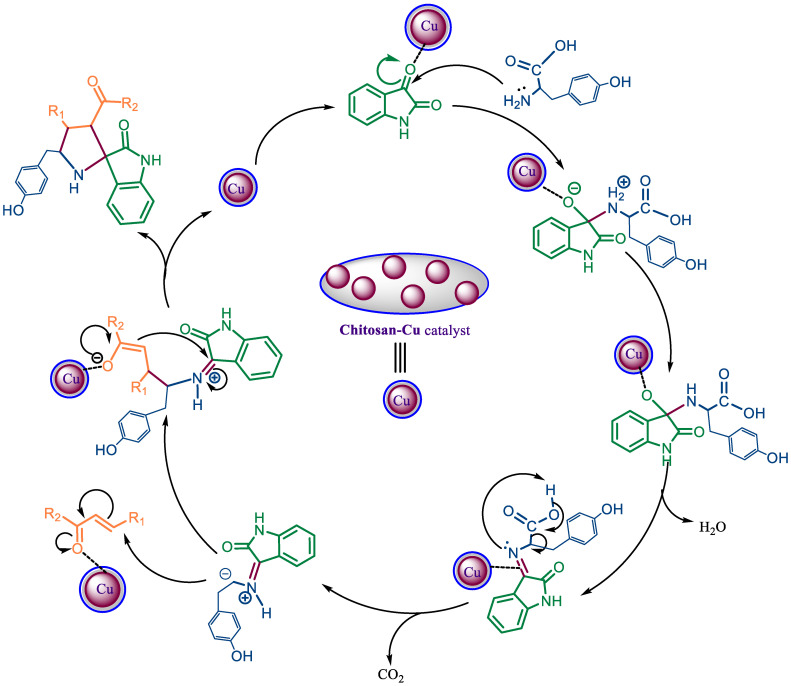
Plausible mechanism for the formation of spiropyrrolidine **3a**.

**Table 1 polymers-15-00429-t001:** Synthesis of spiropyrolidines derivatives (**3a**–**b**).

Compounds	Conventional Method (Refluxing in Methanol)		Solvent-Free Heating Method with Chitosan–Cu Catalyst	
	Time	Yield (%)	Time	Yield (%)
**3a**	5 h	71	15 min	91
**3b**	5 h	65	20 min	88

**Table 2 polymers-15-00429-t002:** Effect of solvents on model reaction.

Entry	Solvents	Temperature	Time	Yield (%)
1	Methanol	Reflux	50 min	75
2	Chloroform	Reflux	1.5 h	64
3	PEG-400	Reflux	1 h	65
4	Water	Reflux	50 min	78
5	Solvent-free	60 °C	15 min	91

**Table 3 polymers-15-00429-t003:** Effect of different catalyst on model reaction.

Entry	Catalysts	Time	Yield
1	Na(CH_3_CO_2_)	80 min	55%
2	Cu(CH_3_CO_2_)_2_	40 min	56%
3	Chitosan	90 min	45%
4	Chitosan–Cu	15 min	91%

**Table 4 polymers-15-00429-t004:** Recycling study of catalyst on the model reaction.

Entry	No. of Cycles	Yield (%)	Time (min)
1	I	91	15
2	II	91	15
3	III	91	15
4	IV	91	15
5	V	88	15

**Table 5 polymers-15-00429-t005:** Structural unique crystal data and refinement parameters for **3a**.

Designed Code	19015
CCDC No.	2215469
Compound’s empirical formula	C_35_H_29_N_2_O_3_BrFe
Formula weight (F. wt.)	661.381
Temp./K	296.15
Crystal system	monoclinic
Space group	*P*2_1_/c
Length of side *a*/Å	16.7910 (13)
Length of side *b*/Å	7.8298 (6)
Length of side *c*/Å	23.1312 (16)
(Alpha Angle) *α*/°	90
(Beta Angle) *β*/°	106.316 (8)
(Gamma angle) *γ*/°	90
Volume/Å^3^	2918.6 (4)
*Z*	4
*ρ*_calc_ g/cm^3^	1.505
*μ*/mm^−1^	1.925
*F*(000)	1353.2
Crystal size/mm^3^	0.42 × 0.35 × 0.33
Radiation used	Mo Kα (*λ* = 0.71073)
2θ/°	5.78 to 58.54 (range for data collection)
Index ranges for collection (*h,k,l*)	−15 ≤ h ≤ 22, −9 ≤ k ≤ 9, −28 ≤ l ≤ 31
Reflections collected	16408
Independent reflections	6913 [*R*_int_ = 0.0379, *R*_sigma_ = 0.0540]
Data	6913
Restraints	0
Parameters	380
Goodness-of-fit on *F*^2^	1.055
Final *R* indexes [I ≥ 2σ (I)]	R_1_ = 0.0453, wR_2_ = 0.1011
Final *R* indexes [all data]	R_1_ = 0.0929, wR_2_ = 0.1279

**Table 6 polymers-15-00429-t006:** Hydrogen bonds for **3a**.

D	H	A	d(D-H)/Å	d(H-A)/Å	d(D-A)/Å	D-H-A/°
O3	H3a	O2 ^1^	0.820	1.833 (4)	2.650 (3)	173.9 (2)
C15	H15	O1 ^2^	0.930	2.486 (5)	3.401 (4)	167.7 (1)
O3	H1O	O2 ^3^	0.820	1.833 (4)	2.650 (3)	173.9 (4)

^1^ 2 − X, −1/2 + Y, 3/2-Z; ^2^ x, y + 1, z and ^3^ −x + 2, y − 1/2, −z + 3/2.

**Table 7 polymers-15-00429-t007:** Antibacterial activity (diameter of zone of inhibition, mm, of compounds **3a**–**b**).

	Diameter of Zone of Inhibition (mm)				
	Gram-Positive Bacteria			Gram-Negative Bacteria	
	*S. pyogenes*	*S. aureus*	*K. pneumoniae*	*P. aeruginosa*	*E. coli*
**3a**	14.4 ± 0.4	13.1 ± 0.5	15.7 ± 0.4	15.2 ± 0.4	14.1 ± 0.5
**3b**	17.4 ± 0.5	16.3 ± 0.4	16.2 ± 0.5	18.1 ± 0.4	17.5 ± 0.5
standard	23.0 ± 0.2	22.0 ± 0.2	19.0 ± 0.2	32.0 ± 0.3	27.0 ± 0.2
DMSO	-	-	-	-	-

Positive control (standard); ciprofloxacin and negative control (DMSO) measured by the halo zone test (unit, mm).

**Table 8 polymers-15-00429-t008:** MIC and MBC results of compounds **3a–b** and positive control ciprofloxacin.

Compounds	Gram-Positive Bacteria						Gram-Negative Bacteria			
	*S. pyogenes*		*MRSA* *		*K. pneumoniae*		*P. aeruginosa*		*E. coli*	
	MIC	MBC	MIC	MBC	MIC	MBC	MIC	MBC	MIC	MBC
**3a**	25	>100	25	100	50	100	50	50	50	50
**3b**	25	100	25	>100	50	>100	100	>100	50	>100
Standard	12.5	12.5	6.25	12.5	6.25	25	12.5	25	6.25	12.5

** Methicillin-resistant Staphylococcus aureus (MRSA+Ve)*.

## Data Availability

Not applicable.

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
