# Peer review of "Chitosan–Cu Catalyzed Novel Ferrocenated Spiropyrrolidines: Green Synthesis, Single Crystal X-ray Diffraction, Hirshfeld Surface and Antibacterial Studies"

_polymers, 2023, doi:10.3390/polym15020429_

Round 1

Reviewer 1 Report

The article authored by Asad et al. describes the use of a material, the result of treating chitosan with copper, in the catalysis of the formation reaction of spiropyrrolidines from tyrosine/glycine and isatin. The activity of the compounds formed against different bacteria was determined and the X-ray structure of one of the compounds was also determined.

In general, the content of the article is interesting and deserves to be published. My main reservation is that the content of the work does not fit the aim and scope of the journal Polymers. The article hardly deals with the possible nature of the chitosan-Cu polymer (reported before) or in what sense its nature can be especially activating the reaction. In fact, the greatest effort developed in the work is the obtaining of two molecular compounds, their characterization and properties. I sincerely recommend that the authors submit this article to another journal dedicated to the synthesis of organic or molecular compounds.

Other aspects to consider:

-          - Summary, line 21, this sentence is confusing. The unit cell of this compound contains exactly four molecules.

-          - The synthesis of the Chitosan-Cu material should be better described or, if it is simply a reproduction of an already described method (as the inclusion of reference 25 in the section title seems to suggest) explicitly stated.

-          - Compounds 3a and 3b seem to be characterized by different spectroscopies. It is a little strange that the determination of its purity was not made by elemental analysis. Given that it will be used to determine its biological activity, can there still be small amounts of Cu?

-          - I do not understand the arguments given by the authors to justify the greater activity of the material in the catalysis of the reaction. Do they suggest that copper supported on chitosan offers a larger catalysis area than copper acetate itself?

Minor points:

-          - Line 300, please include points in the r.m.s. deviation

-          - Describe (in the heading) what is the activity parameter of the compounds included in table 7.

-          - Why do we use OAc as an abbreviation for acetate? Please replace with CH3CO2!

Reviewer 2 Report

The paper proposes a green solvent-free synthesis method using Cu-Chitosan composite. The observed antibacterial properties of 3a–b products is an important result. The authors carefully and in detail studied the structure of compounds.

  Comments:

1. in the abstract, it is necessary to indicate specific compounds, and not use the notation 3a-c

2. The title of section “2.2. Synthesis of catalyst, Chitosan-Cu [25]is confused. What does the reference in the title mean? If a known technique was taken, then this should be indicated in the text. Nothing is said about chitosan, at what stage it was added and in what quantity.

3. on the lines 126-142 the designation of some symbols was missing. Text needs to be corrected

4. Did the amount of Cu in the catalyst was determined?

5. What is the reason of the decrease of yield after the 5 cycle (Table 4)? Is it caused the Cu leaching?

6. The Introduction section should be supplemented by a more specific discussion of the information available in the literature about Сu -Chitosan systems.

Reviewer 3 Report

This paper describes the ferrocene grafted spiropyrrolidine hybrids, which were synthesized by 1-(4-bromophenyl)-ferrocene-prop-2-en-1-one using 1,3-dipolar cycloaddition reaction as dipolarophile with different azomethine ylides via solvent-free reaction using Chitosan-Cu. The spectroscopic techniques were used for establishing the structures of synthesized spiropyrrolidines and further validated by a single crystal X-ray diffraction study, which is added with Hirshfeld surface analysis. The newly synthesized spiropyrrolidine was determined for antibacterial significance, and efficacy compared with a standard drug. In my opinion, I would like to recommend its publication after the following revisions and supplements as shown below:

1)     I would suggest changing the title of subchapter 3.1. Chemistry. It is not precise enough.

2)     It would be more fortunate to wording along the crystallographic direction [ ] instead of along the plane (e.g. line 324). This wording appears several times in the text.

3)     In Table 6, please standardize the notation of bond lengths and angles to the same decimal place. In addition, symmetry references should be written below the table.

4) I am asking for unification in all text: only some words 3a,3b, etc., are bolded (e.g. in line 237, these words are not bold). The same situation applies to the terms: scheme and figure (e.g. line 267).

5)     The caption to Figures 3 and 4 states that the HS analysis and fingerprint plot were performed for compound 1, not 3a. It is not as described. Please check.

6)     The reference to Etter's publication (e.g. M. C. Etter, J. C. MacDonald and J. Bernstein, Acta Crystallogr., Sect. B: Struct. Sci., 1990, 46, 256.) is missing from the text when recital R44(10) is given (line 316). Moreover, this motif is not visible in Figures 2 and S1. I suggest making a new drawing for SI.

7) Sentences on lines 315 to 320 should be reworded. Sentences are incomplete and poorly worded.

8)     Missing % sign on line 339.

Round 2

Reviewer 1 Report

There is an aspect that is argued in the "Cover letter" by the authors that I must comment on. X-ray diffraction is not a suitable method to ensure the purity of a material under any circumstances. When it comes to powder diffraction, we can determine the relative content of different phases, if these have sufficient crystallinity (but it does not report anything that is amorphous). For obvious reasons, when it comes to a single crystal, we only have information about it, but not about the rest of the material that we neglected when preparing the sample. However, I have to remind you that a common practice for the solution of the single crystal structure of proteins is to impurity the crystals with heavy metals. Most of the time the metal is incorporated into the net with little or no modification of the original phase.

In short, I recommend to the authors that in the future, they try to determine the purity of their compounds through a determination of their % C,H,N (S)

Author Response

Dear Reviewer

We are highly grateful to the kind reviewer for keen reviewed the article and good suggestion for future work. We will remember and discuss the elemental analysis in our future work.

Thanks and Regards